# Natural Disasters, Economic Growth, and Carbon Emissions: Empirical Analysis of Chinese Data Based on a Nonlinear Auto-Regressive Distributed Lag Model

**Ming Cao [1], Yiming Xu [2,\*], Yuanhong Sun [2] and Dingbang Cang [2]**

[1] School of Public Policy & Management, China University of Mining and Technology, Xuzhou 221116, China; cm_cumt@126.com
[2] College of Science, North China Institute of Science and Technology, Langfang 065201, China; syh@ncist.edu.cn (Y.S.); cdbjd@163.com (D.C.)
\* Correspondence: xym@ncist.edu.cn

**Abstract:** China has a high frequency of natural disasters and it has become the economy with the largest carbon emissions in recent years. In this study, we mainly investigated the relationships between carbon emissions and natural disaster losses in China, as well as considering important factors such as economic growth and new energy consumption. Time series data for China from 2000 to 2020 were selected and based on the nonlinear auto-regressive distributed lag model method, a short-term error correction model and long-term co-integration relationship model were obtained between carbon emissions and their related factors. The results showed that in the long run, there is a significant nonlinear relationship between carbon emissions, new energy consumption and direct economic losses from natural disasters. There is a significant U-shaped relationship between natural disasters and carbon emissions, that is, natural disaster losses will significantly inhibit carbon emissions before they are below a certain threshold, but fewer natural disaster losses will increase carbon emissions. On the contrary, there is an inverted U-shaped relationship between new energy consumption and carbon emissions. When new energy consumption exceeds a certain threshold, it will help carbon peak early. In the short term, the impact of natural disasters on carbon emissions in the current period is significantly positive and higher than that in the lagged period, but the impact of its square term is negative. The short-term error correction model coefficient is $-0.6467$, and the error will be corrected when the short-term volatility deviates from the long-term equilibrium. These results suggest that attention should be paid to reducing disaster losses and the low-carbon reconstruction path for natural disasters, as well as continuously improving the level of new energy utilization, accelerating the pace of energy substitution, and promoting economic transformation for achieving "carbon peaking" in China.

**Keywords:** carbon emissions; economic growth; natural disasters; new energy

## 1. Introduction

China has complex geographical and climatic conditions, as well as a wide variety of natural disasters, and this country has the highest frequency of natural disasters in the world. Floods, droughts, earthquakes, typhoons, mudslides, forest fires, and other disasters occur every year, and they have obvious regional, seasonal, and phased characteristics. These frequent natural disasters have severe consequences, such as high death tolls, the destruction of infrastructure, and stagnated regional economic development. From 1995 to 2020, the average direct economic losses caused by natural disasters in China each year totaled 300 billion yuan (Figure 1).

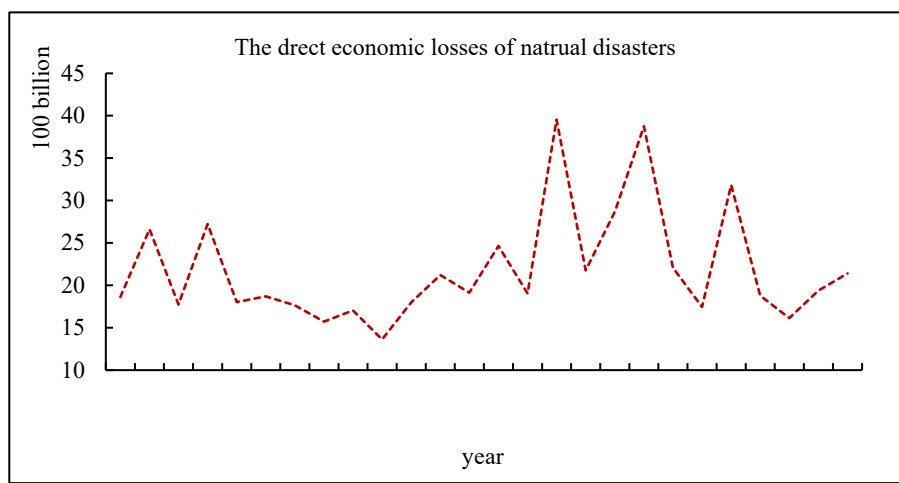

**Figure 1.** The direct economic losses of natural disasters in China. Source: China Statistical Yearbook (1996–2021).

The direct economic losses caused by natural disasters in China were the highest in the world, and this amount accounted for about 1–4% of the total GDP (Gross Domestic Product) each year (Figure 2). Empirical evidence suggests that natural disasters and environmental damage are inextricably linked, where human economic activity is accompanied by damage to the natural environment, which then further exacerbates the occurrence of natural disasters. In particular, challenges such as extreme weather and temperature rises caused by global climate change are important issues that affect the world at present.

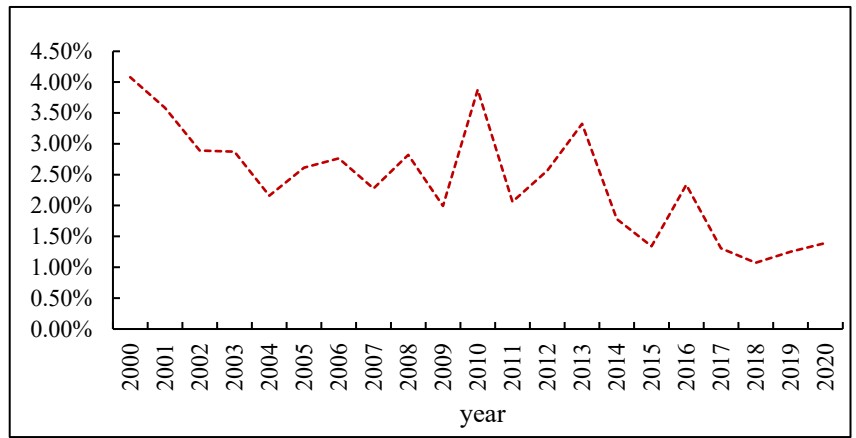

**Figure 2.** Proportion of direct economic losses as GDP caused by disasters. Source: China Statistical Yearbook (2001–2021).

In addition, China is the world's largest energy consumer and its total energy consumption was 4.98 billion tons of standard coal in 2020, where coal consumption accounted for 56.8% of the total energy consumption, and natural gas, hydropower, nuclear power, wind power, and other forms of clean energy consumption accounted for 24.3%. Since 2007, China has surpassed the United States in terms of total carbon emissions and it is ranked first in the world. According to statistics, China's carbon emissions increased each year from 2000 to 2020 (as shown in Figure 3), and the total carbon emissions were nearly 10 billion tons in 2020, with an average annual growth rate of about 5%. At the 75th session of the United Nations General Assembly in September 2020, China clearly proposed "carbon peaking" by 2030 and "carbon neutrality" by 2060, or the "double carbon" target. This plan vigorously seeks to peak carbon emissions in the years before 2030 and achieve carbon neutrality in the years before 2060. Actively optimizing the energy consumption

structure, promoting transformation and upgrading of the industrial structure, and vigorously supporting technological innovation have become important methods for controlling carbon emissions in China.

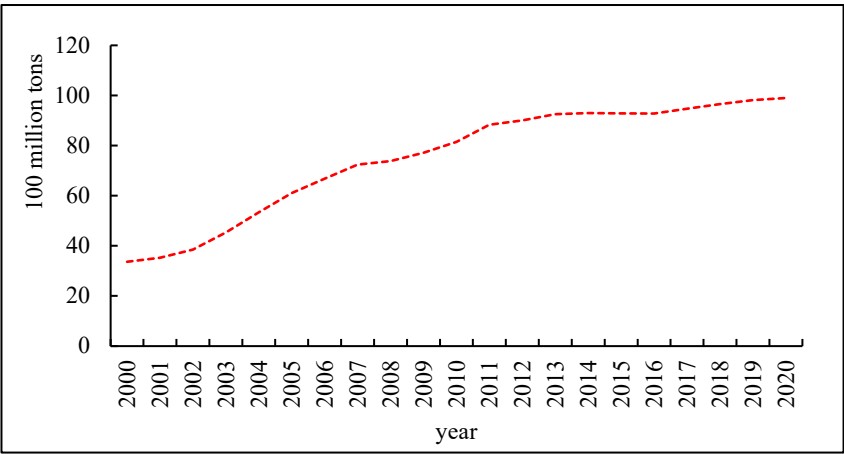

**Figure 3.** China's carbon emissions from 2000 to 2020. Source: BP World Energy Statistical Yearbook (2021).

There are complex interactions between the natural environment and social systems, and the outbreak of COVID-19 once again made the impact of natural disasters a hot topic in academia. In particular, the impacts of natural disasters on social development and economic growth have received widespread attention, but few relevant studies have investigated the impacts of natural disasters on carbon emissions, and thus more empirical tests are still needed. Therefore, in the present study, we investigated the relationships between carbon emissions, economic growth, and natural disasters in China, and determined the dynamic relationships among these variables by empirical analysis in order to suggest suitable policies for mitigating carbon emissions, achieving "carbon peaking", and preventing and reducing disasters. The remainder of this paper is organized as follows. In Section 2, we review recent studies into the effects of natural disasters on economic growth, energy consumption, and carbon emissions. In Section 2, we present our empirical analysis of the long-term and short-term relationships among these variables based on a time-series auto-regressive distributed lag model. In Section 4, we give our conclusions and policy suggestions.

## 2. Literature Review

### 2.1. Relationships between Natural Disasters and Economic Growth

It is well known that human economic activities can damage the natural environment, but people are more concerned about the impacts of natural disasters on economic development. Obviously, natural disasters will have impacts on regional social development, especially major disasters, which can incur huge costs in terms of casualties and property losses, as well as serious consequences such as inflation, financial deficit, and increased poverty, thereby significantly affecting the physical and mental health of people, and economic growth [1–3].

The relationships between economic development and natural disasters have been investigated throughout the world since the mid-1990s, and disaster economics research is currently a hot topic. It is widely accepted that natural disasters have negative impacts on economic growth. In particular, Baig et al. investigated the relationships between natural disasters and economic growth in Pakistan by using the co-integration test method and auto-regressive distributed lag model to show that there was a one-way causal relationship between natural disasters and economic growth in both the short and long term, where natural disasters inhibited economic growth [4]. Kunreuter et al. found that natural disasters lead to large losses and the increased demand for relief materials can crowd out

economic development resources to inhibit economic growth [5]. Noy found that natural disasters adversely affected the economy in the short run, but with regional variability, where developed countries were better able to cope with disaster shocks whereas the consequences of natural disasters were more severe in developing countries [6]. Raddatz also concluded that economic growth in low-income countries was more vulnerable to climate disasters after examining disaster data for 112 countries [7].

In addition, some studies found the opposite relationship between disasters and economic growth, where they argued that disasters could also promote economic growth to a certain extent. Based on case studies of disasters in developing countries, Albala et al. found that the post-disaster economic growth of most countries was rapid, thereby indicating that natural disasters can promote economic development to some extent [8]. Skidemore et al. studied the effects of meteorological and geological disasters on economic growth and found that geological disasters had negative impacts on growth, whereas meteorological disasters made positive contributions to economic growth [9]. Davis and Raschky et al. also found that natural disasters played roles in promoting economic growth [10,11].

### 2.2. Impacts of Natural Disasters on Energy Consumption and Carbon Emissions

Energy is a necessary basic requirement to support economic and social development, and energy consumption, especially fossil energy use, is a major source of carbon emissions. According to statistics, China's carbon emissions from energy activities accounted for about 29% of the total global carbon emissions in 2019 and about 87% of China's total social carbon emissions. Some studies have shown that the impacts of natural disasters on economic development will be transmitted to energy consumption due to the close relationship between the economy and energy [12,13]. Lee et al. studied the relationships between energy and natural disasters in 123 countries and found that natural disasters had negative and significant impacts on the consumption of oil, renewable energy, and nuclear energy and that natural disasters could significantly reduce the demand for energy consumption [14]. Some studies found opposing effects of natural disasters on energy consumption, whereas Doytch et al. found a positive relationship between natural disasters and energy consumption by analyzing panel data from 80 countries [15]. They also found that the impacts of natural disasters on energy consumption varied depending on the economic level, where disasters had positive impacts on renewable energy consumption in high-income economies, positive impacts on industrial energy consumption in low-income economies, and positive impacts on residential energy consumption in middle-income economies. Mishra et al. and Paramati et al. showed that natural disasters caused by climate change can be frequent and that renewable energy sources should replace traditional energy sources (coal, natural gas, and oil) in order to reduce carbon emissions [16,17].

Few studies have examined the impacts of natural disasters on carbon emissions. However, due to the correlation between the economy and energy consumption, it is generally considered that natural disasters will reduce the consumption demand to some extent, and thus restrain carbon emissions [18–21]. Dou et al. used a set of panel data from 138 countries to test the relationships between natural disasters and carbon dioxide emissions through panel quantile regression. Their empirical results showed that natural disasters significantly reduced carbon emissions, where natural disasters directly reduced carbon emissions but also indirectly contributed to carbon reduction by curbing energy consumption, and the level of technology was an important moderator of the relationship between natural disasters and carbon emissions [22]. In the current research literature, there is a lack of reliable conclusions on how natural disasters affect carbon emissions. One explanation is that natural disasters lead to poverty, which in turn reduces energy consumption and ultimately carbon emissions [23,24].

Through the review, we found that there is little literature discussing the impact of natural disasters on $CO_2$ emissions, compared to studies analyzing the impact of natural disasters on economic growth or energy consumption, but this is valuable for assessing

the impact of natural disasters on the environment. Moreover, the explanation for the mediating role of natural disasters in significantly reducing $CO_2$ emissions is not clear, and relevant studies are insufficient. In addition, in this literature, we found almost no empirical studies using a Nonlinear Auto-regressive Distributed Lag Model model to analyze Chinese sample data.

### 3. Models and Data Analysis

#### 3.1. Model Construction

In this study, we analyzed the relationships between natural disasters, economic growth, and carbon emissions. In particular, we constructed an econometric analysis model with carbon dioxide emissions as the explanatory variable, and GDP and natural disasters as the main explanatory variables. According to previous research by Baig [4], Dou [22], Dong [25], and Kasman et al. [26], new energy was also included in the analysis framework, and we defined the multi-factor theoretical model of carbon emission as follows:

$$Carb = f(GDP, Disa, NE) \tag{1}$$

where *Carb* represents carbon emissions, *GDP* represents the scale of economic development, *NE* represents the amount of new energy used, and *Disa* represents the loss value of natural disasters. We performed statistical analysis using the auto-regressive distributed lag model, which is a relatively new co-integration test first proposed by Charemza et al. [27,28]. When studying the time series relationship, this model does not require that each series is co-integrated of the same order, but only that the co-integration order of each time series does not exceed 1. The boundary test is also robust in the case of small samples, and the parameters obtained from the auto-regressive distributed lag model are unbiased and valid when the explanatory variables are endogenous. The steps in auto-regressive distributed lag model analysis mainly include model building, unit root testing, boundary co-integration testing, long-run and short-run coefficient estimation for variables, and stability testing for the model. First, the following log-linear auto-regressive distributed lag model l is established to reduce the absolute value of the data and facilitate the calculation:

$$LnCarb = \alpha_1 + \sum_{i=1}^{k_1} \beta_{1i}(LnGDP)_{t-i} + \sum_{i=1}^{k_2} \beta_{2i}(LnNE)_{t-i} + \sum_{i=1}^{k_3} \beta_{3i}(LnDisa)_{t-i} + \sum_{i=1}^{k_4} \beta_{4i}((LnDisa)*(LnDisa))_{t-i} + \sum_{i=1}^{k_5} \beta_{5i}((LnNE)*(LnNE))_{t-i}$$
$$\gamma_1(LnGDP)_{t-1} + \gamma_2(LnNE)_{t-1} + \gamma_3(LnDisa)_{t-1} + \gamma_4((LnDisa)*(LnDisa))_{t-1} + \gamma_5((LnDisa)*(LnDisa))_{t-1} + \varepsilon_{1t} \tag{2}$$

where $k_i$ denotes the optimal lagged terms, and $\beta_{mi}(m = 1, 2, 3, 4, 5)$ and $\gamma_{mi}$ $(m = 1, 2, 3, 4, 5)$ are the short-run and long-run coefficients, respectively. The existence of a stable long-run relationship between the variables can be determined by using the auto-regressive distributed lag model boundary co-integration test. The hypotheses for the test are as follows.

$$H_0 : \gamma_1 = \gamma_2 = \gamma_3 = \gamma_4 = \gamma_5 = 0$$
$$H_1 : \gamma_1 \neq 0 \text{ or } \gamma_2 \neq 0 \text{ or } \gamma_3 \neq 0 \text{ or } \gamma_4 \neq 0 \text{ or } \gamma_5 \neq 0$$

If the F-statistic is larger than the larger critical value, the original hypothesis of no co-integration relationship can be rejected regardless of whether the series is I(0) or I(1). The following model can be developed to estimate the long-run coefficients:

$$LnCarb = \alpha_2 + \sum_{i=1}^{\delta_1} \eta_{1i}(LnGDP)_{t-i} + \sum_{i=1}^{\delta_2} \eta_{2i}(LnNE)_{t-i} + \sum_{i=1}^{\delta_3} \eta_{3i}(LnDisa)_{t-i} + \sum_{i=1}^{\delta_4} \eta_{4i}((LnDisa)*(LnDisa))_{t-i} + \sum_{i=1}^{\delta_5} \eta_{5i}((LnNE)*(LnNE))_{t-i} + \varepsilon_{2t} \tag{3}$$

where $\delta_i$ is the optimal lag term determined by Akaike's information criterion and $\eta_{mi}(m = 1, 2, 3, 4, 5)$ is the long-term coefficient. The short-term dynamic coefficients can be derived from the auto-regressive distributed lag error correction model as follows:

$$\Delta(LnCarb) = \alpha_3 + \sum_{i=1}^{\xi_1} \rho_{1i}\Delta(LnGDP)_{t-i} + \sum_{i=1}^{\xi_2} \rho_{2i}\Delta(LnNE)_{t-i} + \sum_{i=1}^{\xi_3} \rho_{3i}\Delta(LnDisa)_{t-i} + \sum_{i=1}^{\xi_4} \rho_{4i}\Delta((LnDisa)*L(nDisa))_{t-i} +$$
$$\sum_{i=1}^{\xi_5} \rho_{5i}\Delta((LnNE)*(LnNE))_{t-i} + \lambda ECM_{t-1} + \varepsilon_{3t} \tag{4}$$

where $\Delta$ denotes the first-order difference of the variables, $\rho_{mi}(m = 1, 2, 3, 4, 5)$ denotes the short-term dynamic coefficients, $ECM_{t-1}$ is the error correction term, and $\lambda$ reflects the speed of adjustment to the long term when there is a short-term deviation.

### 3.2. Data Description

To construct the model and analyze the relationships between variables, we obtained a time series data package containing carbon dioxide emissions, *GDP*, new energy consumption and natural disaster losses. In order to facilitate the analysis, the direct economic loss caused by natural disasters was selected as the measure of the degree of natural disaster loss. Data descriptions for each variable are presented in Table 1.

**Table 1.** Descriptions of variables.

| Variable | Data Description |
| --- | --- |
| *LnCarb* | $CO_2$ emissions, calculated by emission factor method (taking natural logarithm) |
| *LnGDP* | *GDP*, calculated at constant prices in 1995 (taking natural logarithm) |
| *LnNE* | Other energy consumption except coal and oil (taking natural logarithm) |
| *LnDisa* | Direct economic loss caused by natural disasters, calculated at constant price in 1995 (taking natural logarithm) |

The data study period ranged from 2000 to 2020, where carbon dioxide emissions data were derived from the BP World Energy Statistical Yearbook 2021, *GDP* and energy consumption; disaster direct economic losses data were taken from China Statistical Yearbook 2000–2021; and *GDP* and *Disa* data were at the 1995 constant price calculation. Statistical descriptions of the variables are presented in Table 2.

**Table 2.** Statistical descriptions of variables.

| | *LnCarb* | *LnGDP* | *LnNE* | *LnDisa* |
| --- | --- | --- | --- | --- |
| Mean | 8.7141 | 11.7554 | 10.3491 | 7.7028 |
| Median | 8.8968 | 11.8336 | 10.3894 | 7.5541 |
| Maximum | 9.2002 | 12.4919 | 11.2795 | 9.0965 |
| Minimum | 8.0159 | 10.8225 | 9.2806 | 7.2178 |
| Std. Dev. | 0.4560 | 0.5478 | 0.6185 | 0.3889 |
| Skewness | −0.1057 | −0.2730 | 0.3260 | 1.9939 |
| Kurtosis | 1.8000 | 1.7602 | 2.2924 | 7.4683 |

### 3.3. Unit Root and Boundary Co-Integration Testing

Non-stationary time series data can lead to pseudo-regression [29], so we used the augmented Dicky–Fuller (referred to as stationary in the following) test to detect the stationarity of the data. This test basically aims to differentiate the test series and determine the series by comparing the correlations in the differenced series with the original series, and whether there is a unit root. The test results are shown in Table 3.

**Table 3.** Augmented Dicky–Fuller test results.

| Variable | Sequence | 5% | *t*-test | Probability |
| --- | --- | --- | --- | --- |
| *LnCarb* | level | −0.8439 | −3.6584 | 0.9433 |
| | 1st difference | −3.4249 * | −3.1199 | 0.0298 |
| *LnGDP* | level | −3.7105 | 0.3454 | 0.9970 |
| | 1st difference | −3.7912 * | −4.3777 | 0.0196 |
| *LnNE* | level | −3.6584 | −4.6518 | 0.0740 |
| | 1st difference | −3.6736 ** | −9.1330 | 0.0001 |
| *LnDisa* | level | −3.6032 | −4.9087 | 0.0031 |

* indicates a significant difference at the 5% probability level, ** indicates a significant difference at the 1% probability level.

The results obtained using the augmented Dicky–Fuller test showed that all variables were I(0) or I(1) series, thereby satisfying the preconditions for further boundary co-integration tests proposed by Pesaran [29]. We determined the optimal lag order of 2 according to Akaike's information criterion and obtained the boundary co-integration test results for the model as shown in Table 4.

**Table 4.** Bounds F-tests for co-integration.

| Significance | 10% | | 5% | | 1% | |
|---|---|---|---|---|---|---|
| | **I(0)** | **I(1)** | **I(0)** | **I(1)** | **I(0)** | **I(1)** |
| Critical value | 1.81 | 2.93 | 2.14 | 3.34 | 2.82 | 4.21 |
| F-Test Statistics | 11.7323 ** | | | | | |

** indicates a significant difference at the 1% probability level.

The test results in Table 4 show that the F statistic for the model was greater than the maximum critical values of 2.82 and 4.21 at the 1% significance level. Thus, the test was passed and there was a long-term stable co-integration relationship between the test variables; therefore, short-term and long-term parameter estimation could be conducted for the constructed auto-regressive distributed lag model.

*3.4. Nonlinear Auto-Regressive Distributed Lag Model Estimation*

Based on the sample data of China from 2000 to 2020, a long-term model between variables is obtained, and the parameter estimation results are shown in Table 5.

**Table 5.** Estimations of co-integration relationships.

| Variable | Coefficient | Standard Error | *t*-Test | Probability |
|---|---|---|---|---|
| *LnGDP* | 1.2132 | 0.2871 | 4.2251 | 0.0012 |
| *LnNE* | 2.5634 | 1.0393 | 2.4666 | 0.0297 |
| *LnNE*$^2$ | −0.1503 | 0.0530 | −2.8372 | 0.0150 |
| *LnDisa* | −3.8990 | 1.3161 | −2.9625 | 0.0119 |
| *LnDisa*$^2$ | 0.2404 | 0.0763 | 3.1499 | 0.0084 |

Table 5 shows that in the long term, Economic growth is a major factor in carbon emissions, and the faster the *GDP* growth rate, the greater the carbon emissions. There was a significant nonlinear relationship between carbon emission, new energy consumption and direct economic losses from natural disasters. The coefficient of the secondary term of new energy consumption is negative, indicating that there is an inverted U-shaped relationship between the two, which shows that only when the new energy consumption exceeds a certain threshold can the carbon emission be inhibited.

The coefficient for the secondary term of natural disaster losses was positive, and thus there was a significant U-shaped relationship between natural disasters and carbon emissions. A low frequency of natural disasters reduced carbon emissions, whereas a large frequency of natural disasters promotes carbon emissions. It is possible that when disaster losses did not exceed a certain threshold value, the consumption by residents, transportation, and other areas was affected to a certain extent, thereby reducing carbon emissions. When the disaster losses exceeded a certain threshold, large-scale post-disaster reconstruction promoted the level of social consumption to increase carbon emissions.

For comparison with the linear model, the long-term estimation results obtained by the linear auto-regressive distributed lag model are shown in Table 6, which indicates that the estimation results were not significant. The nonlinear auto-regressive distributed lag model was more suitable for analyzing the relationships between natural disaster losses and carbon emissions than the linear model, and different frequencies of natural disaster losses had different effects on carbon emissions.

**Table 6.** Linear model estimation results.

| Variable | Coefficient | Standard Error | *t*-Test | Probability |
|----------|-------------|----------------|----------|-------------|
| *LnGDP* | 0.5296 | 0.2746 | 1.9290 | 0.0743 |
| *LnNE* | 0.1581 | 0.0951 | 1.6619 | 0.1187 |
| *LnDisa* | 0.1181 | 0.2476 | 0.4770 | 0.6407 |

Next, the parameter estimation results obtained by the short-term error correction model are shown in Table 7.

**Table 7.** Error correction model estimation results.

| Variable | Coefficient | Standard Error | *t*-Test | Probability |
|----------|-------------|----------------|----------|-------------|
| D($LnCarb(-1)$) | 0.8767 | 0.0497 | 17.6239 | 0.0000 |
| D($LnDisa$) | 1.0541 | 0.1333 | 7.9069 | 0.0002 |
| D($LnDisa(-1)$) | 0.5333 | 0.1187 | 4.4948 | 0.0041 |
| D($LnDisa^2$) | $-0.0659$ | 0.0082 | $-7.9989$ | 0.0002 |
| D($LnDisa \cdot LnDisa(-1)$) | $-0.0333$ | 0.0073 | $-4.5603$ | 0.0038 |
| D($LnGDP$) | 0.2001 | 0.1452 | 1.3776 | 0.2175 |
| D($LnGDP(-1)$) | 1.3884 | 0.2470 | 5.6211 | 0.0014 |
| D($LnNE$) | $-1.9970$ | 0.3564 | $-5.6030$ | 0.0014 |
| D($LnNE(-1)$) | $-2.7561$ | 0.3754 | $-7.3420$ | 0.0003 |
| D($LnNE^2$) | 0.0973 | 0.0186 | 5.2270 | 0.0020 |
| D($LnNE \cdot LnNE(-1)$) | 0.1456 | 0.0193 | 7.5655 | 0.0003 |
| $CointEq(-1)$ | $-0.6468$ | 0.0483 | $-13.3892$ | 0.0000 |
| $R^2$ | 0.9999 | | | |

Table 7 shows that both new energy sources and their lagged periods had suppressive effects on carbon emissions in the short term, thereby indicating that the use of new energy sources had a sustained effect on reducing carbon emissions, but the lagged effect was higher than the current period value. The effects of the current and lagged *GDP* on carbon emissions were promoting, where, but the former was less than the latter. The effect of natural disasters on carbon emissions was significantly positive in the current period, which was higher than that in the lagged period. However, in the short term, the squared term of disaster has a negative impact on carbon emissions, and the effect of a delay period is greater than that of the current period. The short-term error correction model coefficient for the model was –0.6467, and thus it was significant at the 5% level, and it adjusted at a rate of approximately 64.7% when short-term fluctuations deviated from the long-run equilibrium.

### 3.5. Granger Causality Testing

According to Engle and Granger's causality test [30], a time series was applied to analyze the causal relationships between carbon emissions and associated variables, thereby determining whether "x caused y by testing whether the current y could be explained by the past x (whether the inclusion of lagged values of x enhanced the explanatory power)." The test results are shown in Table 8.

**Table 8.** Granger non-causality test results.

| Null Hypothesis | F-Statistics | Probability |
|-----------------|--------------|-------------|
| *LnNE* is not the Granger Cause of *LnCarb* | 3.6322 | 0.0461 |
| *LnGDP* is not the Granger Cause of *LnNE* | 3.6602 | 0.0452 |
| *LnDisa* is not the Granger Cause of *LnNE* | 2.4772 | 0.0986 |

According to the test results, *LnNE* was the Granger Cause of *LnCarb* at the 5% test level, and *LnGDP* and *LnDisa* were the Granger Cause of *LnNE* at the 5% and 10% test level, respectively. So *LnGDP* or *LnDisa* may be the indirect Granger Cause of *LnCarb*.

### 3.6. Model Diagnosis

Based on similar studies [31,32], the diagnostic results for the model are presented in Table 9, where the residual Q statistic showed that the model was not serially correlated, the Jarque–Bera test statistic showed that the residuals were normal, the Autoregressive Conditional Heteroskedasticity (ARCH) test showed that there was no heteroskedasticity, and the Ramsey test indicated that the model was reliable.

To avoid an unreliable model due to the instability of the parameters, the model stability was tested based on the recursive cumulative sum of residuals (CUSUM) and recursive cumulative sum of squared residuals (CUSUMSQ), and the test results are shown in Figures 4 and 5.

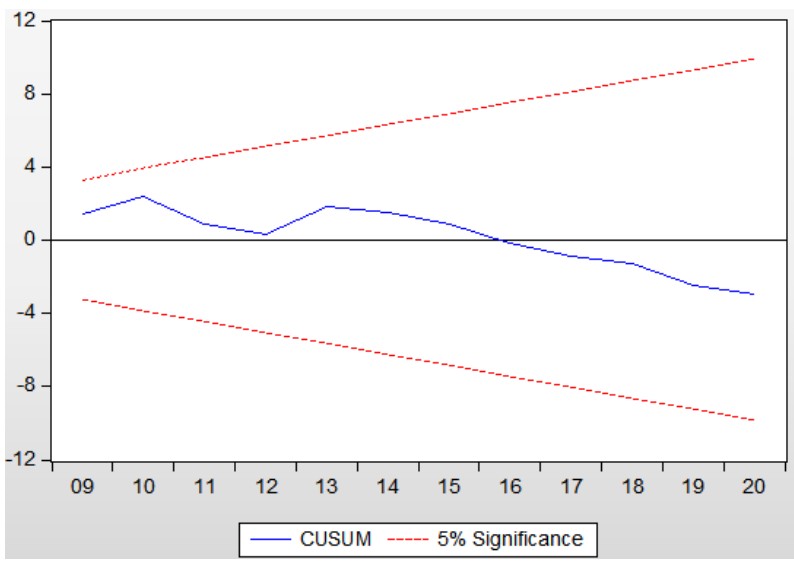

**Figure 4.** Recursive cumulative sum of residuals (CUSUM) test results.

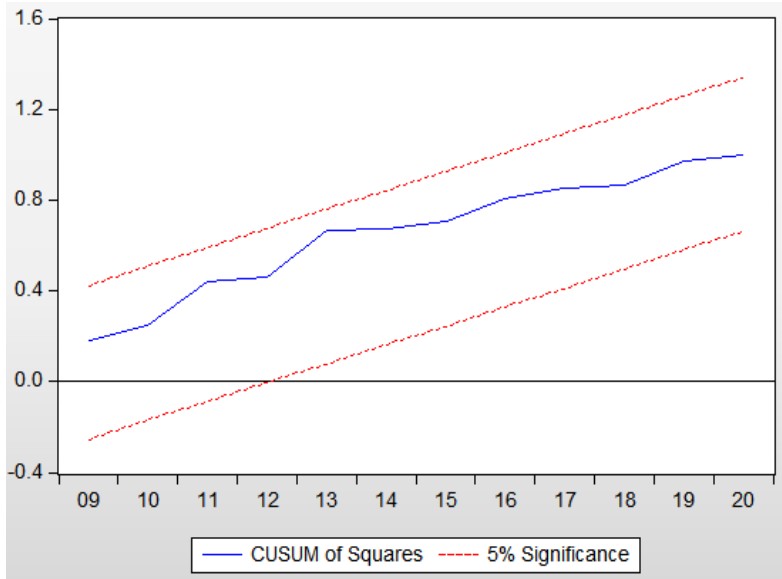

**Figure 5.** Recursive cumulative sum of squared residuals (CUSUMSQ) test results.

The two lines in Figures 4 and 5 represent the upper and lower critical values at a significance level of 5%, and the broken lines in the middle represent the CUSUM and CUSUMSQ values over time. The CUSUM and CUSUMSQ values calculated with *LnCarb*

as the dependent variable were located between the upper and lower critical values. Thus, the model was generally effective and the research conclusions were reliable.

**Table 9.** Model diagnosis results.

|  | *t*-Test | Probability |
|---|---|---|
| Ramsey RESET test | 1.3414 | 0.2068 |
| Jarque-Bera test | 2.8532 | 0.2413 |
| ARCH test | 0.0189 | 0.8919 |

## 4. Conclusions and Discussion

To investigate the relationships between natural disasters and carbon emissions, we used time series data for China from 2000 to 2020 and included economic growth and new energy consumption factors. The augmented Dicky–Fuller test results for the sample data indicated that all variables were stationary at the horizontal state or first-order difference series, and thus suitable for further econometric analysis. The short-run error correction model and long-run co-integration model between *LnCarb* and related variables *LnGDP*, *LnNE*, and *LnDisa* were obtained using the nonlinear auto-regressive distributed lag model method. Granger causality testing was conducted to analyze the causal relationships between carbon emissions and other factors. We find that there is a significant nonlinear relationship between carbon emission, new energy consumption and direct economic losses from natural disasters. Economic growth is a major factor in increasing carbon emissions. There is a significant U-shaped relationship between natural disasters and carbon emissions and an inverted U-shaped relationship between new energy consumption and carbon emissions. This shows that when the loss of natural disasters is below a certain threshold, carbon emissions will be significantly inhibited, while when the consumption of new energy exceeds a certain threshold, it will help reduce carbon emissions. The short-term error correction model coefficient is −0.6467, and the error will be corrected when the short-term fluctuation deviates from the long-term equilibrium. Based on the main conclusions and findings presented above, the following recommendations are made regarding the formulation of carbon emissions reduction policies in China.

Natural disasters can significantly promote carbon emissions, attention should be paid to identifying, preventing, and mitigating natural disasters by preparing scientific summaries and predictions based on the spatial and temporal evolution characteristics of the occurrence of various types of natural disasters, as well as continuously strengthening the construction of emergency management systems and building an effective emergency response mechanism to reduce the impacts of natural disasters on social development. However, it should be noted that in the reconstruction process after natural disaster losses, the rebound in carbon emissions due to inefficient construction should be avoided, and opportunities should be exploited to optimize low-carbon reconstruction paths according to the local conditions.

In any case, in order to reduce carbon emissions, we should resolutely and continuously improve the level of new energy use, and new energy substitution is a strategic path for achieving "carbon peaking" in China. Accelerating the pace of energy replacement and increasing the proportion of new energy consumption can help to protect the environment and better resist climate change to reduce the frequency of meteorological disasters. In practice, it is necessary to provide a better market environment for the development and utilization of new energy, improve the energy technology and industrial innovation system, accelerate research into key core technologies and equipment in the energy field, promote major breakthroughs in green and low-carbon technologies, and improve the efficiency of new energy utilization.

Finally, economic transformation and high-quality development are fundamental for achieving "carbon peaking" in China. China is a developing country and it is necessary to develop its economy and improve living standards for people but under constraints in terms of resource depletion and the environmental carrying capacity. Thus, the only

strategy in the future is high-quality development via transformation and developing a low-carbon society and low-carbon economy. China is a major trading country in the world and foreign trade is an important driving force for China's economic development, but the expansion of export trade incurs huge environmental resource costs, so it is necessary to keep export-oriented foreign trade in a reasonable range, change the economic growth mode through technological innovation, reduce the dependence of economic growth on trade, focus on green foreign trade, optimize the commodity structure of foreign trade, and reduce the carbon emissions of products.

**Author Contributions:** Conceptualization, M.C. and D.C.; methodology, Y.X. and D.C.; software, Y.X.; validation, Y.X., Y.S. and M.C.; data curation, Y.X., Y.S. and M.C.; writing—review and editing, M.C. and Y.X.; funding acquisition, M.C. All authors have read and agreed to the published version of the manuscript.

**Funding:** This research was funded by the 2020 Hebei Social Science Development Research Project (No. 20200202045) and 2022 Langfang Science and Technology Project "Analysis of the relationship between disaster losses and carbon emissions in China".

**Institutional Review Board Statement:** Not applicable.

**Informed Consent Statement:** Not applicable.

**Data Availability Statement:** The evaluation data selected in this article are all derived from "China Statistical Yearbook" (1996–2020) and BP World Energy Statistical Yearbook (2021).

**Conflicts of Interest:** The authors declare no conflict of interest.

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
