# Peer review of "Natural Disasters, Economic Growth, and Carbon Emissions: Empirical Analysis of Chinese Data Based on a Nonlinear Auto-Regressive Distributed Lag Model"

_sustainability, doi:10.3390/su152115210_

Round 1

Reviewer 1 Report

The manuscript entitled “Natural Disasters, Economic Growth, and Carbon Emissions: Empirical Analysis of Chinese Data Based on a Nonlinear Auto-regressive Distributed Lag Model” is reviewed.

·         The authors argue that they used the NARDL test, but they did not report positive and negative shocks in the long run

·         The authors should provide the pattern of the natural disaster dataset with an explanation in the introduction section.

·         21 years do not enough to run the NARDL test with four independent variables, the authors should use quarterly data or should transform the dataset from annual to quarterly using cubic or quadratic approaches.  CUSUMSQ and CUSUM clearly show the lack of a dataset in the figure with 3-year x-axis. 

There is a minor grammatical errors in the text

Author Response

Dear reviewer, thank you for your guidance. We have tried our best to revise the paper according to your comments. Please check the uploaded attachment, which is the explanation of the revision of the paper, and allow us to submit the revised manuscript.

Reviewer 2 Report

The paper studied the relationship between carbon emissions economic growth, and natural disaster losses in China, and the research was conducted with a wealth of reliable data for China from 2000 to 2020. And a series of models were carried out to explore the relationship. However, there is not much difference between the conclusions drawn and public recognition after complex calculations and analysis. In other words, where is the significance of the above work reflected? So, the paper could be published after some modifications. In preparing the revision, the novelty of the work as well as the significance of the approach used in the work was encouraged to address.

No.

Author Response

(The authors gave the same response as above.)

Reviewer 3 Report

Comments:

1- The author should minimize the abstract by focusing on the key findings and policy suggestion, instead of mentioning all the results.

2- The introduction is mainly about the fact and figures, instead of highlighting the problem statement, contribution and objective of the study.

3- At the end of literature section, the author should provide the research gap which should be determined by the previous literature. Moreover, few of the relevant studies are missing which need to be cited, such as:

(2023). Empirical Evidence of Environmental Technologies , Renewable Energy and Tourism to Minimize the Environmental Damages : Implication of Advanced Panel Analysis. International Journal of Environmental Research and Public Health, 20(6), 5118.

(2023). The Significance of Governance Indicators to Achieve Carbon Neutrality: A New Insight of Life Expectancy. Sustainability, 15(1), 1–20. https://doi.org/10.3390/su15010766

(2022). Empirical relationship between creativity and carbon intensity: a case of OPEC countries. Environmental Science and Pollution Research, 30(13), 38886–38897. https://doi.org/10.1007/S11356-022-24903-8/TABLES/7

(2022). The Nexus of Energy, Green Economy, Blue Economy, and Carbon Neutrality Targets. Energies, 15(18), 6767. https://doi.org/10.3390/EN15186767

(2022). Investigate solutions to mitigate CO2 emissions: the case of China. Journal of Environmental Planning and Management, 65(11), 2054–2080. https://doi.org/10.1080/09640568.2021.1952859

(2022). Impact of Urbanization and expansion of forest investment to mitigates CO2 emissions in China. Weather, Climate, and Society, 14(3), 681–696. https://doi.org/10.1175/WCAS-D-21-0101.1

(2021). Effects of Forestry on Carbon Emissions in China : Evidence From a Dynamic Spatial Durbin Model. Frontiers in Environmental Science, 9(October), 1–15. https://doi.org/10.3389/fenvs.2021.760675

(2020). Revisiting the empirical relationship among the main targets of sustainable development: Growth, education, health and carbon emissions. Sustainable Development, 29(2), 419–440. https://doi.org/10.1002/SD.2156

(2019). Economic and non-economic sector reforms in carbon mitigation: Empirical evidence from Chinese provinces. Structural Change and Economic Dynamics, 49. https://doi.org/10.1016/j.strueco.2019.01.003

(2019). Nexus Among Economic Growth, Education, Health, and Environment: Dynamic Analysis of World-Level Data. Frontiers in Public Health, 7(October), 1–15. https://doi.org/10.3389/fpubh.2019.00307

4- Theoretical reasoning is missing for this study.

5- The author should apply more unit root tests, instead of relying on single test.

6- As all the variables are stationary at first difference, why the author choose bond cointegration? Why Kao or Pedroni are not suitable?

Author Response

(The authors gave the same response as above.)

Reviewer 4 Report

Dear authors

My comments on your manuscript are presented as follows:

1- In the reference section, author should add new references related to the subject of the project which have been recently published by the scholars.

2- The results illustrated in the manuscript should be compared with the results presented in the previous studies conducted by the scholars.

3- The lack-of-fit the model used in this research should be illustrated by the authors in the manuscript.

4- The importance of the study presented in the manuscript should be highlighted in the introduction section.

5- The main differences between this study and other studies conducted by the scholars should be presented in the literature review.

6- The future trend for following this research should be illustrated in the conclusion section.   

Author Response

(The authors gave the same response as above.)

Round 2

Reviewer 1 Report

The revised study can be accepted 

Reviewer 3 Report

No further comments

Reviewer 4 Report

Dear authors

My comments on your manuscript have been conducted.